# Short-Term Adverse Effects Following Booster Dose of Inactivated-Virus vs. Adenoviral-Vector COVID-19 Vaccines in Algeria: A Cross-Sectional Study of the General Population

**DOI:** 10.3390/vaccines10111781

**Published:** 2022-10-22

**Authors:** Mohamed Lounis, Hani Amir Aouissi, Samir Abdelhadi, Mohammed Amir Rais, Salem Belkessa, Djihad Bencherit

**Affiliations:** 1Department of Agro-Veterinary Science, Faculty of Natural and Life Sciences, University of Ziane Achour, Djelfa 17000, Algeria; 2Scientific and Technical Research Center on Arid Regions (CRSTRA), Biskra 07000, Algeria; 3Laboratoire de Recherche et d’Étude en Aménagement et Urbanisme (LREAU), Université des Sciences et de la Technologie (USTHB), Algiers 16000, Algeria; 4Environmental Research Center (CRE), Badji-Mokhtar Annaba University, Annaba 23000, Algeria; 5Department of Mathematics, Faculty of Exact Sciences, Frères Mentouri University, Constantine 25000, Algeria; 6Department of Dentistry, Faculty of Medicine, University of Algiers Benyoucef Benkhedda, Algiers 16000, Algeria; 7Department of Biology, Faculty of Natural and Life Sciences, University of Ziane Achour, Djelfa 17000, Algeria

**Keywords:** COVID-19, vaccines, booster, side effects, adenoviral-vector vaccines, inactivated-virus vaccines, Algeria

## Abstract

COVID-19 booster vaccines have been adopted in almost all countries to enhance the immune response and combat the emergence of new variants. Algeria adopted this strategy in November 2021. This study was conducted to consider the self-reported side effects of COVID-19 booster vaccines by Algerians who were vaccinated with a booster dose of one of the approved inactivated-virus vaccines, such as BBIBP-CorV and CoronaVac, or one of the adenoviral-vector-based vaccines, such as Gam-COVID-Vac, AZD1222 and Ad26.COV2.S, and to determine the eventual risk factors. A cross-sectional study using an online self-administered questionnaire (SAQ) was conducted in Algeria between 28 April 2022, and 20 July 2022. A descriptive analysis of the 196 individuals who were included showed a nearly equal distribution of adenoviral- (52%) and inactivated-virus vaccines (48%) and of males (49.5%) and females (50.5%). The results showed that 74.7% of the studied population reported at least one local or systemic side effect. These side effects were more frequent among adenoviral-vector vaccinees (87.3%) than inactivated-virus vaccinees (60.6%) (sig. < 0.001). Injection site pain (40.3%), heat at the injection site (21.4%), and arm pain (16.3%) were the most common local side effects. These signs generally appeared in the first 12 h (73.3%) and generally lasted less than 24 h (32.8%). More interestingly, these signs differed from those that followed the administration of primer doses (48.5%) and were generally more severe (37%). The same observation was reported for systemic side effects, where the signs were especially most severe in the adenoviral-vaccinated group (49.4% vs. 20.8%; sig. = 0.001). These signs generally appeared within the first day (63.6%) and mostly disappeared before two days (50.8%), with fatigue (41.8%), fever (41.3%), and headache (30.1%) being the most common. Adenoviral-vector vaccinees (62.7%) were more likely to use medications to manage these side effects than were inactivated-virus vaccinees (45.7%) (sig. = 0.035) and paracetamol (48.5%) was the most used medication. Adenoviral-based vaccines were the types of vaccines that were most likely to cause side effects. In addition, being female increased the risk of developing side effects; regular medication was associated with local side effects among inactivated-virus vaccinees; and previous infection with COVID-19 was associated with systemic and local side effects among adenovirus-based vaccinees. These results support the short-term safety of booster vaccines, as has been reported for primer doses.

## 1. Introduction

In late 2019, the world witnessed the emergence of one of the most noteworthy health threats of the 21st century, the COVID-19 pandemic [1]. As of 12 September 2022, the total number of infections worldwide was estimated to be more than 605 million and nearly 6.5 million deaths were recorded [2]. The disease has also had a range of disastrous effects on the economic sector, due to the preventive measures that were applied during the first months of the pandemic, which included border closings, social distancing, and quarantine [3,4].

Following multiple non-pharmaceutical measures, efforts were directed toward pharmaceutical measures by accelerating research trials to find the best vaccine for this disease. These efforts were concluded with the development of various types of vaccines in just one year [5]. In fact, six COVID-19 vaccines were urgently approved by the World Health Organization (WHO) in or after December 2020. Thus, a vaccination campaign race began in different countries. Later, other types of vaccines were also used in different countries. Currently, there are 11 vaccines approved by the WHO and, in at least one country, 47 vaccines have been approved [6]. These vaccines are of multiple types, including inactivated-virus, vector-based, protein-based, and mRNA-based vaccines [7].

The safety and effectiveness of the primer doses of these vaccines have been proved [8]. In fact, safety data about the two primer doses from previous studies showed that, as with any other drug, these vaccines could induce some short and mild adverse effects, including flu-like symptoms (e.g., headache, fatigue, and myalgia) and injection-site reactions [4,9,10,11,12,13,14,15]. Some severe signs, such as anaphylaxis, coagulation, myocarditis, thyroiditis, and even death, have also been reported, albeit rarely [16].

Regarding the efficacy of these vaccines, previous studies have confirmed that the vaccines are substantially associated with a decrease in hospitalization and death rates [4]. However, multiple studies have shown that, as with other vaccines, the efficacy of the COVID-19 vaccines decreases a few months after the second primer dose [17,18,19].

The waning protection of these vaccines, in association with the emergence of multiple variants of SARS-CoV-2 that were less sensitive than the first variants to these vaccines, has led several countries to recommend a booster dose for their populations, with either homologous or heterologous vaccines [16,20,21]. Consequently, the phenomenon of booster hesitancy/rejection emerged, although its frequency has not been as high as it was for primer doses.

In general, the phenomenon of vaccine hesitancy and rejection is mainly related to vaccines’ efficacy and safety. For instance, the effectiveness of boosters has been proven, in clinical trials and/or after their homologation, in the enhancement of immune responses, the control of infections, and the treatment of severe cases of the disease [21,22,23,24]. However, results regarding booster side effects are scarce.

In its vaccination campaign, Algeria approved five vaccines, including inactivated-virus vaccines (BBIBP-CorV and CoronaVac) and adenoviral-vector-based vaccines (Gam-COVID-Vac, AZD1222, and Ad26.COV2.S). The number of vaccinated people in Algeria with at least one dose since January 2021 has reached 7.84 million (17.75% of the total population), while the number of fully vaccinated people according to the initial protocol is estimated at 6.48 individuals (14.67% of the total population) [25]. The campaign for booster vaccination started in November 2021. However, despite the encouraging results regarding its acceptance, as reported in our previous study [26], the number of individuals who received a COVID-19 vaccine booster did not exceed 514,063, which represents approximately 1.3% of the total population [27].

Therefore, this study aimed to investigate the short-term side effects of the administration of the COVID-19 vaccine booster dose among the general population in Algeria and the associated risk factors. To the best of our knowledge, this is the first study conducted in Algeria about the side effects of COVID-19 vaccine boosters.

## 2. Materials and Methods

### 2.1. Study Design

The current cross-sectional study was conducted via a self-administered questionnaire (SAQ) with the aim of describing and evaluating the self-reported side effects of COVID-19 vaccines among Algerian people who have been vaccinated with a booster dose of one of the approved vaccines.

To achieve these objectives, the survey was carried out between April 28, 2022, and July 20, 2022. The designed SAQ was administered online, using Google Forms (Google LLC, Menlo Park, CA, USA, 2021). It was disseminated through social media platforms and emails, using uniform resource locator (URL), and a quick response (QR) was sought.

### 2.2. Participants

A convenient online sampling was used in the current study. The target population of this study was Algerian citizens living in Algeria who received a booster dose of one of the approved COVID-19 vaccines. Participants who received inactivated-virus or adenoviral-vector vaccines in their primers and booster doses were included. Participants who received other types of vaccines (i.e., other than inactivated-virus or adenoviral-vector vaccines) in their primer doses were excluded.

The minimum sample size was calculated using the following equation [28]:$$n=Z^{2}\frac{pq}{e^{2}}$$

In the equation, *n*: is the minimum sample size (here, *n* = 196);
Z^2^ = 1.96 for α = 0.05 (confidence interval of 95%);*p* is the proportion (*p* = 0.5; in the 2022 study of Lounis et al. [14], approximately 50% of the vaccinated population developed at least one local or systemic side effect);*q* = 1 − *p*; and*e* is the accepted margin of error (ME) (here, we chose an ME of 7%, or 0.07);

In total, 208 responses were received in this study. However, only 196 participants were included for further analyses. Participation in the survey was voluntary and there were no incentives or compensation. No identity data were requested from the participants and anonymity was ensured to control Hawthorne’s effect and information bias.

### 2.3. Instrument

The SAQ utilized in this study was obtained and adapted from previous studies regarding COVID-19 primer and booster doses [9,10,11,12,13,14,15,20,21]. The questionnaire was translated and distributed in the Arabic and French languages.

In this SAQ, 40 multiple-choice items were used, which were grouped into three categories: (i) demographic features that included sex, age, and professional status; (ii) anamnestic features, including chronic diseases, BMI, ABO group, regular used medications, allergies, smoking status, previous COVID-19 infection, COVID-19 vaccine primer and booster type, the duration between primer and booster doses, and the participant’s state of mind during vaccination; and (iii) post-vaccination local and systemic side effects, their onset and duration, post-vaccination-used medications, and hospitalization.

### 2.4. Ethical Considerations

This study protocol was reviewed and approved by the Scientific Committee of the Faculty of Natural and Life Sciences, University of Djelfa, on 25 April 2022. (It was signed by the president of the scientific committee and the dean of the faculty).

Electronic informed consent was obtained from all participants before their enrolment in the study. Participants could not continue the survey if they declined consent. They could leave the survey at any time, in accordance with the Declaration of Helsinki’s ethical principles.

### 2.5. Statistical Analysis

Data were statistically analyzed utilizing SPSS version 22.0 (SPSS Inc.: Chicago, IL, USA, 2013). Data were presented as frequencies (*n*) and percentages (%) to summarize nominal and ordinal data. The association between independent and dependent variables was evaluated using Chi-squared (χ^2^) and Fisher tests.

Subsequently, multinomial logistic regression was used to evaluate the suggested risk factors of post-vaccination side effects following boosters with either of the two types of vaccines. All data were analyzed with a confidence level (CI) of 95% and a significance level (sig.) of ≤ 0.05.

## 3. Results

### 3.1. Demographic and Anamnestic Characteristics of the Study Population

In the current study, 208 individuals vaccinated with one of the approved vaccines (adenoviral vaccines or inactivated-virus vaccines) and receiving a booster dose from different regions of Algeria completed the questionnaire during the study period from 28 April 2022, to 20 July 2022.

Due to incomplete responses or other reasons (i.e., individuals who received different types of vaccines in their primer doses), 12 questionnaires were excluded. A total of 196 responses were included in this study.

Demographic characteristics showed that the participants were nearly equally distributed in terms of sex (male, *n* = 97 vs. female, *n* = 99). With respect to age, the most reported categories were for those participants who were aged between 18 and 30 years (65, 33.2%) and between 31 and 40 years (*n* = 55, 28.2%). For BMI, weight, and blood group, the figures were as follows: normal BMI (*n* = 92, 46.9%); overweight (*n* = 73, 37.8%); O blood group (*n* = 80, 40.8%).

Regarding health status, most of the respondents (*n* = 164, 83.7%) were non-smokers and 37.2% (*n* = 73) suffered from an allergy.

Further, forty-four of the participants (22.4%) suffered from at least one chronic disease, with hypertension (9.2%, *n* = 18), diabetes (8.2%, *n* = 16), and respiratory diseases (4.1%, *n* = 8) as the most common. Regarding regular medication, 15.3% of the participants (*n* = 30) were taking antihistamines and 13.8% (*n* = 27) used antibiotics regularly (Table 1).

### 3.2. COVID-19 Infection History and Vaccine Primer and Booster Characteristics

Concerning COVID-19 history, the majority of the respondents (71.9%, *n* = 141) had contracted the disease, and most of them were infected one time (80.1%, *n* = 113) and infected generally before primer vaccination (86.5%, *n* = 122).

For the primer doses of vaccination, vaccines were distributed among inactivated-virus (60.7%, *n* = 117) and adenoviral-vector-based (39.3%, *n* = 77) vaccines, with Sinovac and AstraZeneca as the most represented, respectively.

On the other hand, the adenoviral-vector-based (52%, *n* = 102) vaccines were used more for a booster dose than were inactivated-virus vaccines (48%, *n* = 94); the Janssen vaccine was the most used among the first type.

Most of the participants were vaccinated (71.4%, *n* = 140) more than 3 months before the survey, and the duration between the second primer and the booster doses was generally between 3 and 6 months (46.9%, *n* = 92).

Overall, 43.3% (*n* = 85) and 40.3% (*n* = 79) of the participants declared that they were serene or indifferent just before the booster vaccination, while 9.2% (*n* = 18) and 6.1% (*n* = 12) were worried or anxious, respectively (Table 2).

### 3.3. Booster Side Effects

Based on our results, 146 individuals (74.7%) reported at least a local or a systemic side effect, with a significant difference between the adenoviral-vector vaccine group (87.3%) and the inactivated-virus vaccine group (60.6%) (sig. < 0.001). In addition, 123 (62.8%) and 125 (63.8%) individuals developed at least one local or systemic side effect, respectively, with a statistical difference between the adenoviral-based vaccines and the inactivated-based vaccines.

#### 3.3.1. Local Side Effects

Our results showed that the most commonly reported local side effects were injection-site pain (40.3%, *n* = 79), heat at the injection site (21.4%, *n* = 42), and arm pain (16.3%, *n* = 32). Redness (12.8%, *n* = 25), injection-site itching and indurations (8.7%, *n* = 17), and swelling (6.1%, *n* = 12) were also reported (Figure 1).

These signs generally appeared generally in the first 12 h (73.7%) and generally lasted less than 24 h (32.8%) or between 24 and 72 h (36.1%). In addition, nearly half of the respondents (48.5%) declared that these signs differed from those of the primer doses; 37% of the respondents declared that these signs were more severe than those that appeared during the primer doses. No statistically significant difference was reported between the two types of vaccines (Table 3).

#### 3.3.2. Systemic Side Effects

With regard to systemic side effects, the most common reported signs were fatigue (41.8%, *n* = 82), fever (41.3%, *n* = 81), and headache (30.1%, *n* = 59). Fatigue and fever in association with chills and arthralgia (which were reported among 21.9% and 21.4% of the respondents, respectively) were significantly more frequent among adenovirus vaccines than inactivated-virus vaccines. Dizziness (17.9%, *n* = 35) and sweating (17.3%, *n* = 34) were also frequent, while other signs, including somnolence, cough, insomnia, dyspnea, nasal discharge, chest pain, abdominal pain, diarrhea, loss of taste/smell, halitosis, and myalgia, were reported with low frequencies varying from 0.5 to 9.2%. Their frequencies were not statistically different among the two vaccinated groups (Figure 2).

Regarding the onset of systemic side effects, most of them appeared within the first day (63.6%), with no difference between the two groups. Systemic side effects appeared more quickly in inactivated-virus vaccinee group than in the adenovirus-based vaccinee group.

Regarding the duration of systemic side effects, they mostly disappeared within two days (50.8%) or between two days and one week (28.3%), with no statistically significant difference between the two groups. These signs were different from those reported during the primer dose vaccination (48.6%), an observation that was significantly more frequent for the adenoviral-vaccinated group (56%) than for the inactivated-virus group (37%) (sig. = 0.032). Regarding the severity of systemic side effects, they were generally more severe than the effects of the primer dose for the adenoviral-vaccinated group (49.4% vs. 20.8%; sig. = 0.001) and less severe than those of the primer dose for the inactivated-vaccine group (41.7% vs. 26%, sig. = 0.003) (Table 4).

Regarding the participants’ ways of managing these post-vaccination adverse effects, more than half (54.6%, *n* = 107) reported that they used medication after vaccination. Medications were more frequently used by adenoviral-vector vaccinees (62.7%, *n* = 62) than by inactivated-virus vaccinees (45.7%, *n* = 43) (sig. = 0.035). Paracetamol (48.5%), poly-vitamins (21.4%, *n* = 42), and herbal traditional infusions (19.4%, *n* = 36) were the most used medications. In addition, 36 individuals declared that they were absent from work after vaccination (mostly those among the adenoviral-vector vaccinees (69.44%, *n* = 25)) for a period varying from 1 to 32 days, while only one person who received inactivated-virus vaccine required hospitalization.

### 3.4. Risk Factors of the Side Effects of Vaccines

In the current study, multiple factors were associated with local and/or systemic side effects. Being female was highly associated with side effects (81.8% vs. 67%) for both local (70.7% vs. 54.6%) and systemic side effects (73.7% vs. 53.6%). In addition, individuals aged more than 50 years (77.1%) and obese persons (80.6%) were more exposed to systemic side effects than were their counterparts, while persons with normal BMI were most significantly associated with a low frequency of any side effects. In contrast, individuals suffering from chronic illnesses were significantly associated with systemic and local side effects (88.6%) and only systemic side effects (77.3%), but not with local side effects. On the other hand, taking medication regularly was significantly associated with local side effects (77.1%) or local and systemic side effects (83.3%), but not with systemic side effects alone.

Another highly associated factor with all side effects (87.3% vs. 60.6%), local side effects (75.5% vs. 51.1%) and systemic side effects (77.5% vs. 46.8%) was the use of the adenoviral-based vaccines, compared with the inactivated-virus vaccines. In addition, the use of adenovirus-based vaccines in primer–booster doses was associated with high frequencies of total, local, and systemic side effects, while the use of inactivated-virus vaccines was associated with the lowest frequencies.

In this study, local and/or systemic side effects were not statistically associated with ABO groups, allergy, smoking, or previous COVID-19 infections (Table 5).

The results of the logistic multinomial regression showed that one of the most determinant factors for developing side effects is the type of vaccine. In fact, the adenoviral-based vaccines are highly associated with the emergence of side effects (OR: 4.520, CI 95%: 2.017–10.127), either for local (OR: 3.648, CI 95%: 1.832–7.266) or systemic side effects (OR: 2.995, CI 95%: 1.517–5.911). In addition, being female is a determinant factor in developing side effects (OR: 2.714, CI 95%: 1.179–6.25), either for local (OR: 2.17, CI 95%: 1.027–4.585) or systemic side effects (OR: 2.956, CI 95%: 1.416–6.173). In addition, taking medication was associated with the presence of local side effects, especially among inactivated-virus vaccinees (OR: 6.37, CI 95%: 1.96–20.833), while previous infection with COVID-19 was associated with systemic and local side effects (OR: 5.983, CI 95%: 1.009–35.5), especially for adenovirus-based vaccinees (OR: 6.369, CI: 1.432–28.336).

## 4. Discussion

The current study aimed to determine the self-reported adverse effects and possible associated risk factors among Algerian people who receive a booster dose of approved vaccines. These side effects were compared between adenoviral-based vaccinees and inactivated-virus vaccinees. In fact, describing the side effects of booster vaccines could help to confirm the safety of these vaccines and, therefore, help to reduce the phenomenon of vaccine (booster) hesitancy, which is a phenomenon described for both primer and booster doses of COVID-19 in Algeria [26,29] and around the world [30,31]. Only limited data are available around the world about the side effects of boosters [16,20,32,33,34,35,36]. Moreover, little is known about the frequency of these side effects in Algeria.

In this study, 74.7% of the surveyed population reported at least a local or a systemic side effect. These results concurred with those of Nguyen et al. [16], who reported at least one side effect in 79.1% of the Vietnamese population after the booster dose. Additionally, in our study, 62.8% and 63.8% of the respondents developed at least one local or systemic side effect, respectively. These side effects were more frequent in adenoviral-vector vaccinees (87.3%) than in inactivated-virus vaccinees (60.6%). These findings confirm the previous reports of multiple studies that reported that side effects were less frequent for inactivated-virus vaccines than for the other types (adenoviral-based vaccines and mRNA-based vaccines) after both primer doses [14,16] and booster doses [35,36].

In this work, nearly half of the participants (48.5%) declared that the side effects they developed after the booster dose were different, and generally more severe (37.7%), than those that developed after the primer doses. This finding was particularly observed by individuals who received adenoviral-based vaccines, and more specifically by those who were vaccinated with inactivated-virus vaccines for their primer doses (i.e., those who mixed inactivated-virus and adenoviral-based vaccines in the primer–booster doses, respectively); nearly three quarters of the respondents (73.7%) declared that systemic side effects were more severe after the booster dose. Moreover, half of those respondents who mixed adenoviral-based and inactivated-virus (primer–booster) vaccines, respectively, declared that the side effects were less severe. These results confirmed the severity of the side effects of adenoviral-based vaccines, compared with inactivated-virus vaccines, as reported in our previous study on primer doses in Algeria [14]. However, these signs remain manageable and generally disappear after a few days.

On the other hand, recently, Chansaenroj et al. [32] reported that the side effects that follow mixed inactivated-viral vector and inactivated-mRNA (primer–booster) doses are generally mild. They confirmed that the reactogenicity after this protocol was acceptable. Other heterologous programs were also shown to induce a greater immune response than that of the homologous protocol [37]. Similarly, the results of multiple studies reported that side effects were the same or milder after the booster dose, compared with the primer doses [16,38,39,40]. However, Rzymski et al. [20] reported that individuals who developed more side effects after the primer doses were more likely to develop side effects after the booster dose. They explained these findings by differences in individual reactions and by the nocebo phenomenon related to anxiety that is associated with misconceptions about potential side effects [20]. In our study, the participants’ states of mind during vaccination were not associated with the appearance of side effects.

Our results showed that injection-site pain (40.3%), heat at the injection site (21.4%), and arm pain (16.3%) were the most common local side effects, while the most frequent systemic side effects were fatigue (41.8%), fever (41.3%), and headache (30.1%). These side effects were generally more frequent in the adenoviral-based vaccinated group than in the inactivated-virus vaccinated group. These results accord with the available findings about booster side effects [16,33,34,35,36,37], with some differences in their frequencies. These side effects were also common after primer doses [9,10,11,12,13,14,15]. They are, however, reported with higher frequency by healthcare workers than were reported after the primer dose, as noted in our previous study [14]. These side effects disappeared quickly and were generally manageable. In fact, only one individual was hospitalized, while 54.6% used medication, and paracetamol was the most used (48.5%). This approach is common, as reported in previous studies [14,15,20]. In this way, previous studies about COVID-19 and other vaccines (such as hepatitis vaccine) reported that the use of paracetamol is safe and did not affect the immune reaction [41,42]. Consequently, this drug has been recommended by some health authorities to relieve the adverse effects of COVID-19 vaccines [43].

Regarding risk factors associated with the development of side effects, our study showed that females, individuals taking regular medications, and those respondents with previous COVID-19 infections were the most exposed to developing local or systemic side effects. The predisposition of females to the side effects of vaccines was previously documented for both primer and booster doses [14,20,36], regardless of the type of vaccine.

The explanation of this phenomenon is related to the behavioral, hormonal, and genetic differences between males and females that lead to different immunological reactions [44]. Based on COVID-19 history, the increasing of side effects could be related to the higher antibody titer after the vaccination of individuals who had past contact with SARS-CoV-2 [14]. The disparate results in multiple studies on multiple vaccines could, however, be related to the fact that the period between COVID-19 infection and vaccination has rarely been determined, which was the case in our study.

In this study, a statistically significant relationship was observed between previous use of medication and local side effects (OR: 3.41, CI 95%: 1.55–7.46). These results, however, need to be interpreted with some caution, due to the various types of medications used and their various mechanisms of action.

Surprisingly, in our study, age, chronic diseases, and BMI were not associated with the development of side effects. In fact, elderly individuals and those with chronic diseases and high BMI are known to be characterized by weak immune-system reactions, confirming previous results of multiple studies that reported that people in these categories developed adverse effects less frequently than those in other age groups [45,46,47,48].

### 4.1. Limitations

Our study had some limitations that could affect our results. The main limitation was the low number of participants, which was related to the low number of individuals who receive vaccine booster doses in Algeria. A second limitation is that the period of study coincided with the return to the “new normal life” and, accordingly, people are generally tending to disregard subjects related to the COVID-19 pandemic. Another limitation was related to the online nature of this survey and the convenient sampling method, which made it difficult to state generalizations about the results. Nevertheless, this method had its benefits, as it mainly covered all regions of Algeria and had a nearly equal distribution regarding sex, age, and vaccine type. Finally, this study described the self-reported side effects of the general population using anonymous responses. These side effects could be confusing for individuals with no medical background. Moreover, the side effects were not verified or confirmed, nor were they officially recorded or documented; they could have been under- or over-described by the participants.

### 4.2. Strengths

To our knowledge, this study is one of the first studies in Algeria, the Middle East, and African countries that investigated the self-reported side effects of COVID-19 booster doses. Therefore, the results of this study could be very helpful in understanding the side effects of vaccines and, accordingly, contribute to the confirmation of the short-term safety of vaccines and to combatting the phenomenon of booster vaccine reluctance, especially in low- and middle-income countries.

## 5. Conclusions

Despite some limitations related to the nature and size of the sample, this study has confirmed the short-term safety of adenoviral-based and inactivated-virus booster doses used in Algeria. In fact, the side effects following such boosters—which included injection site pains and some systemic side effects, such as fever and fatigue—were mainly mild, of short duration, and less severe than those that followed the primer doses, except for individuals who received inactivated-virus or adenoviral-based vaccines for their primer–booster doses.

These safety results could be helpful for the public health authorities in their battle against vaccine hesitancy in general and, in particular, the new phenomenon of booster-dose hesitancy.

## Figures and Tables

**Figure 1 vaccines-10-01781-f001:**
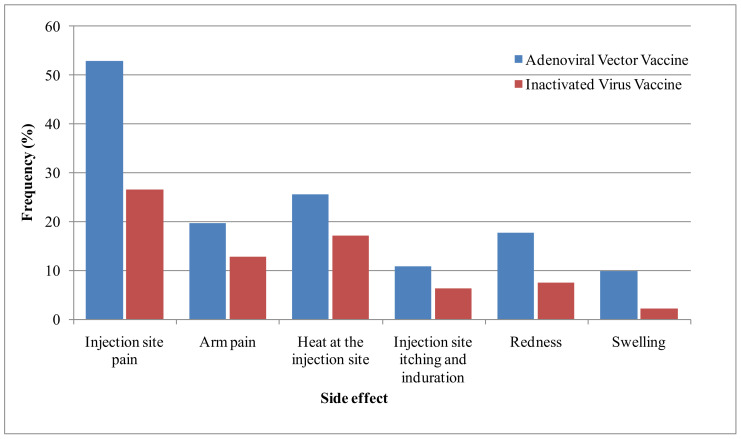
Frequency of local side effects reported by the studied population.

**Figure 2 vaccines-10-01781-f002:**
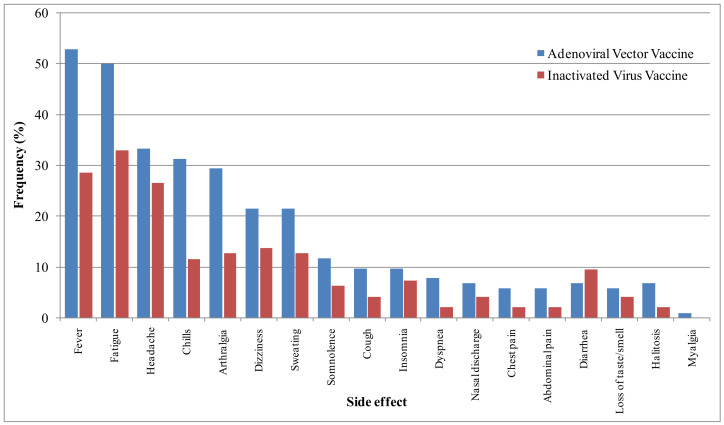
Frequency of systemic side effects of the studied population.

**Table 1 vaccines-10-01781-t001:** Demographic and anamnestic characteristics of the study population.

Variable		Adenoviral-Virus Vaccines (*n* = 102)	Inactivated-Virus Vaccines (*n* = 94)	Total (*n* = 196)
		Number (%)	Number (%)	Number (%)
Age	18–30 years old	30 (29.4)	35 (37.2)	65 (33.2)
	31–40 years old	29 (28.4)	25 (26.6)	54 (27.6)
	41–50 years old	22 (21.6)	20 (21.3)	42 (21.4)
	Over 50 years old	21 (20.6)	14 (14.9)	35 (17.9)
Sex	Male	48 (47.1)	49 (52.1)	97 (49.5)
	Female	54 (52.9)	45 (47.9)	99 (50.5)
ABO group (*n* = 194)	A	30 (29.4)	30 (31.9)	60 (30.6)
	B	13 (12.7)	3 (3.2)	16 (8.2)
	AB	19 (18.6)	19 (20.2)	38 (19.4)
	O	39 (38.2)	41 (43.6)	80 (40.8)
BMI	Normal	45 (44.1)	47 (50)	92 (46.9)
	Obese	38 (37.3)	35 (37.2)	73 (37.2)
	Over Obese	19 (18.6)	12 (12.8)	31 (15.8)
Allergy	(Yes)	38 (37.3)	35 (37.2)	73 (37.2)
Smokers	(Yes)	13 (12.7)	19 (20.2	32 (16.3
Chronic diseases	(Yes)	28 (27.5)	16 (17)	44 (22.4)
Type	Hypertension	9 (8.8)	9 (9.6)	18 (9.2)
	Diabetis	10 (9.8)	6 (6.4)	16 (8.2)
	COPD	6 (5.9)	2 (2.1)	8 (4.1)
	Cardiovascular	2 (2)	2 (2.1)	4 (2.0)
	Thyroid	4 (3.9)	0 (0)	4 (2.0)
	Others	5 (4.9)	6 (6.4)	11 (5.6)
Medication	(Yes)	58 (56.9)	38 (40.4)	96 (49)
Type	Antihistamines	19 (18.6)	11 (11.7)	30 (15.3)
	Antibiotics	17 (16.7)	10 (10.6)	27 (13.8)
	Antihypertensive	8 (7.8)	9 (9.6)	17 (8.7)
	Anti-reflux	9 (8.8)	5 (5.3)	14 (7.1)
	Anti-diabetes	7 (6.9)	5 (5.3)	12 (6.1)
	Anti-sthma	8 (7.8)	3 (3.2)	10 (5.1)
	Contraceptives	6 (5.9)	2 (2.1)	8 (4.1)
	Antdepressants	3 (2.9)	4 (4.3)	7 (3.6)
	Thyroid hormones	4 (3.9)	3 (3.2)	7 (3.6)
	Corticoids	3 (2.9)	4 (4.3)	7 (3.6)
	Cholesterol lowering	3 (2.9)	1 (1.1)	4 (2)
	Anticoagulant	1 (1)	1 (1.1)	2 (1)

**Table 2 vaccines-10-01781-t002:** COVID-19 infection history and booster uptake of the studied population.

Variable		Number	Frequency (%)
COVID-19 infection	No	55	28.1
	Yes	141	71.9
Number of infections	One time	113	80.1
	Two times	23	16.3
	Three times	4	2.8
Onset	Before vaccination	122	86.5
	After the first dose	19	13.5
	After the second dose	35	24.8
	After the third dose	27	19.1
Vaccination timing	From a month to 3 months ago	39	19.9
	From a week to one month ago	12	6.1
	Less than a week ago	5	2.6
	More than 3 months ago	140	71.4
Vaccine first dose	AstraZeneeca (ChAdOx1, AZD1222)	37	18.9
	Janssen (Ad26.CoV2.S)	5	2.6
	Sinopharm (BBIBP-CorV)	20	10.2
	Sinovac (CoronaVac)	99	50.5
	Sputnik V (Gam-COVID-Vac)	35	17.9
Vaccine second dose	AstraZeneeca (ChAdOx1, AZD1222)	36	18.4
	Janssen (Ad26.CoV2.S)	5	2.6
	Sinopharm (BBIBP-CorV)	23	11.7
	Sinovac (CoronaVac)	96	49.0
	Sputnik V (Gam-COVID-Vac)	36	18.4
Vaccine booster	AstraZeneeca (ChAdOx1, AZD1222)	28	14.3
	Janssen (Ad26.CoV2.S)	71	36.2
	Sinopharm (BBIBP-CorV)	13	6.6
	Sinovac (CoronaVac)	81	41.3
	Sputnik V (Gam-COVID-Vac)	3	1.5
Primer doses type	Adenoviral-vector virus	77	39.3
	Inactivated virus	119	60.7
Booster dose type	Adenoviral-vector virus	102	52.0
	Inactivated virus	94	48.0
Time between primer and booster doses (months)	1 month	19	9.7
	1 to 3 months	33	16.8
	3 to 6 months	92	46.9
	More than 6 months	51	26.0
State of mind during booster vaccination	Anxious	12	6.1
	Worried	18	9.2
	Indifferent	79	40.3
	Serene	85	43.4

**Table 3 vaccines-10-01781-t003:** Frequency and characteristics of local side effects of the Algerian population.

Variable		Booster Vaccine Type			Sig.
		Adenoviral-Vector-Virus	Inactivated-Virus	Total	
Local side effects	Injection-site pain	54 (52.9)	25 (26.6)	70 (40.3)	**0.000**
	Arm pain	20 (19.6)	12 (12.8)	32 (16.3)	0.195
	Heat at the injection site	26 (25.5)	16 (17)	42 (21.4)	0.149
	Injection-site itching and induration	11 (10.8)	6 (6.4)	17 (8.7)	0.274 *
	Redness	18 (17.6)	7 (7.4)	25 (12.8)	**0.032**
	Swelling	10 (9.8)	3 (2.1)	12 (6.1)	0.052 *
Onset	<12 h	58 (75.3)	29 (70.7)	87 (73.7)	0.589
	>12 h	10 (13)	10 (24.4)	20 (16.9)	0.129 *
	>24 h	9 (11.7)	2 (4.9)	11 (9.3)	0.327 *
Duration	Less than 24 h	27 (34.2)	12 (30)	39 (32.8)	0.647
	From 24 to 72 h	30 (38)	13 (32.5)	43 (36.1)	0.557
	From 3 days to a week	15(19)	7 (17.5)	22 (18.5)	1 *
	More than a week	7 (8.9)	8 (20)	15 (12.6)	0.141 *
Similarity	Not the same signs	45 (53.6)	21 (40.4)	66 (48.5)	0.135
	The same signs as the first dose	19 (22.6)	15 (28.8)	34 (25)	0.415
	The same signs as the second dose	2 (2.4)	5 (9.6)	7 (5.1)	0.106 *
	The same signs as the two primer doses	19 (21.4)	11 (21.2)	29 (21.3)	0.970
Severity	The same severity as the two primer doses	20 (25.6)	19 (38.8)	39 (30.7)	0.118
	Less severe than the two primer doses	24 (30.8)	17 (34.7)	41 (32.3)	0.645
	More severe than the two primer doses	34 (43.6)	13 (26.5)	47 (37)	0.052

Chi-squared (χ^2^) and Fisher (*) tests were used with a significance level (sig.) of 0.05. Bold character refers to statistically significant values.

**Table 4 vaccines-10-01781-t004:** Frequency and characteristics of systemic side effects of the Algerian population.

Variable	Outcomes	Booster Vaccine Type			Sig.
		Adenoviral-Vector Virus	Inactivated Virus	Total	
Systemic side effects	Fever	54 (52.9)	27 (28.7)	81 (41.3)	**0.001**
	Fatigue	51 (50)	31 (33)	82 (41.8)	**0.016**
	Headache	34 (33.3)	25 (26.6)	59 (30.1)	0.304
	Chills	32 (31.4)	11 (11.7)	43 (21.9)	**0.001**
	Arthralgia	30 (29.4)	12 (12.8)	42 (21.4)	**0.005**
	Dizziness	22 (21.6)	13 (13.8)	35 (17.9)	0.158
	Sweating	12 (21.6)	12 (12.8)	34 (17.3)	0.104
	Somnolence	12 (11.8)	6 (6.4)	18 (9.2)	0.223 *
	Cough	10 (9.8)	4 (4.3)	14 (7.1)	0.169 *
	Insomnia	10 (9.8)	7 (7.4)	17 (8.7)	0.619 *
	Dyspnea	8 (7.8)	2 (2.1)	10 5.1)	0.103 *
	Nasal discharge	7 (6.9)	4 (4.3)	11 (5.6)	0.541 *
	Chest pain	6 (5.9)	2 (2.1)	8 (4.1)	0.282 *
	Abdominal pain	6 (5.9)	2 (2.1)	8 (4.1)	0.282 *
	Diarrhea	7 (6.9)	9 (9.6)	16 (8.2)	0.604 *
	Loss of taste/smell	6 (5.9)	4 (4.3)	10 (5.1)	0.75 *
	Halitosis	7 (6.9)	2 (2.1)	9 (4.6)	0.173 *
	Myalgia	1 (1)	0 (0)	0 (0.5)	1 *
Onset	Immediately	5 (6.6)	8 (17.8)	13 (10.7)	0.071 *
	The first day	52 (68.4)	25 (55.6)	77 (63.6)	0.155
	The first week	12 (15.8)	8 (17.8)	20 (16.5)	0.804 *
	The second week	3 (3.9)	2 (4.4)	5 (4.1)	1 *
	After the first month	4 (5.3)	2 (4.4)	6 (5)	1 *
Duration	Less than 2 days	41 (54.7)	20 (44.4)	61 (50.8)	0.278
	From 2 days to a week	18 (24)	16 (35.6)	34 (28.3)	0.174
	From a week to 2 weeks	7 (9.3)	2 (4.4)	9 (7.5)	0.481 *
	From 2 weeks to 4 weeks	2 (2.7)	1 (2.2)	3 (2.5)	1 *
	More than 4 weeks	7 (9.3)	6 (13.3)	13 (10.8)	0.551 *
Similarity	Not the same signs	47 (56)	21 (37.5)	68 (48.6)	**0.032**
	The same as the first dose	21 (25)	18 (32.1)	39 (27.9)	0.356
	The same as the second dose	3 (3.6)	7 (12.5)	10 (7.1)	0.089 *
	The same as the primer doses	13 (15.5)	10 (17.9)	23 (16.4)	0.817 *
Severity	The same severity as the primer doses	19 (24.7)	18 (37.5)	37 (29.6)	0.127
	Less severe than the primer doses	20 (26)	20 (41.7)	40 (32)	0.067
	More severe than the primer doses	38 (49.4)	10 (20.8)	48 (38.4)	**0.001**

Chi-squared (χ^2^) and Fisher (*) tests were used with a significance level (sig.) of 0.05. Bold character refers to statistically significant values.

**Table 5 vaccines-10-01781-t005:** Risk factors associated with side effects in the Algerian population.

Variable		Local Side Effects		Systemic Side Effects		Total Side Effects	
		Number (%)	Sig.	Number (%)	Sig.	Number (%)	Sig.
Age	18–30 years old	39 (60)	0.574	37 (56.9)	0.160	44 (67.7)	0.124
	31–40 years old	32 (59.3)	0.532	36 (66.7)	0.604	39 (72.2)	0.653
	41–50 years old	29 (69)	0.341	25 (59.5)	0.518	32 (76.2)	0.775
	Over 50 years old	23 (65.7)	0.689	27 (77.1)	**0.036**	31 (88.6)	0.114
Sex	Male	53 (54.6)	**0.02**	52 (53.6)	**0.003**	65 (67)	**0.017**
	Female	70 (70.7)		73 (73.7)		81 (81.8)	
ABO group	A	37 (61.7)	0.703	40 (66.7)	0.528	45 (75)	0.869
	AB	10 (62.5)	1	10 (62.5)	1	11 (68.8)	0.563
	B	27 (71.1)	0.245	27 (71.1)	0.275	31 (81.6)	0.304
	O	48 (60)	0.486	46 (57.5)	0.153	57 (71.3)	0.427
BMI class	Normal	52 (56.5)	0.090	54 (58.7)	0.164	61 (66.3)	**0.013**
	Obese	47 (64.4)	0.716	46 (63)	0.864	58 (79.5)	0.220
	Overly obese	24 (77.4)	0.072	25 (80.6)	**0.041**	27 (87.1)	0.114
Chronic diseases	Yes	32 (72.7)	0.12	34 (77.3)	**0.034**	39 (88.6)	**0.015**
	No	91 (59.9)		91 (59.9)		107 (70.4)	
Allergy	Yes	49 (67.1)	0.33	50 (68.5)	0.290	57 (78.1)	0.374
	No	74 (60.2)		75 (61)		89 (72.4)	
Smoking	Yes	19 (59.4)	0.665	18 (56.3)	0.333	21 (65.6)	0.374
	No	104 (63.4)		107 (65.2)		125 (76.2)	
Medication	Yes	74 (77.1)	**0.000**	67 (69.8)	0.086	80 (83.3)	**0.005**
	No	49 (49)		58 (58)		66 (66)	
COVID-19 infection	Yes	71 (58.7)	0.134	77 (63.6)	0.959	88 (73.8)	0.707
	No	52 (69.3)		48 (64)		58 (76.4)	
Booster type	Adenoviral-virus	79 (77.5)	**0.000**	77 (75.5)	**0.000**	89 (87.3)	**0.000**
	Inactivated-virus	44 (46.8)		48 (51.1)		57 (60.6)	
Mixing/matching (primers-booster)	AA ^+^	43 (79.6)	**0.003**	40 (74.1)	**0.064**	50 (92.6)	**0.000**
	AI ^+^	11 (47.8)	0.115	11 (47.8)	0.09	13 (56.5)	**0.035**
	IA ^+^	38 (74.5)	**0.044**	39 (76.5)	**0.028**	40 (81.6)	0.256
	II ^+^	31 (45.6)	**0.000**	35 (51.5)	**0.009**	43 (61.4)	**0.002**
Time between primers and booster doses (months)	1 month or less	11 (57.9)	0.619	12 (63.2)	1.000	14 (73.7)	1
	1 to 3 months	19 (57.6)	0.472	19 (57.6)	0.431	21 (63.6)	0.122
	3 to 6 months	64 (69.6)	0.076	62 (67.4)	0.297	74 (80.4)	**0.066**
	More than 6 mths	29 (56.9)	0.285	31 (60.8)	0.628	36 (70.6)	0.473

Chi-squared (χ^2^) test was used with a significance level (sig.) of 0.05. Bold character refers to statistically significant values. ^+^ A: adenoviral-based vaccine, I: inactivated-virus vaccine.

## Data Availability

The data that support the findings of this study are available from the corresponding author upon reasonable request.

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
