# Peer review of "Short-Term Adverse Effects Following Booster Dose of Inactivated-Virus vs. Adenoviral-Vector COVID-19 Vaccines in Algeria: A Cross-Sectional Study of the General Population"

_vaccines, 2022, doi:10.3390/vaccines10111781_

Round 1

Reviewer 1 Report

The paper "Short-term Adverse Effects Following Booster Dose of Inactivated Virus vs. Adenoviral Vector COVID-19 Vaccines in Algeria: A Cross-sectional Study Among the General Population" describes the result of a study performed on a sample of Algerian population to characterise the side effects after the booster doses administration.

A self-administred questionnaire has been proposed to a quite uniform sample of the population, receiving about 200 answers (196 used).

The analysis of the questionnaires has been carried out in an almost exaustive way, showing, describing and discussing the results obtained.

The work is good. Maybe a greater number of individuals could be involved; moreover, some additional characteristic can be usefully incuded, for example the geographical distribution (large city, small city, rural areas).

It is suggested a full text check for the correction of the English form. For example, pag. 3, line 119, "side effects" repeated, or the paragraph starting from line 308 of page 12, the "were" at line 309 seems wrong.

Author Response

Dear Reviewer,

First of all, we would like to express our sincere gratitude for the time dedicated to the review this paper. We highly appreciate your detailed and constructive comments, which allowed us to improve the manuscript.

The suggestions were greatly helpful for us, and we addressed all comments in the revised paper. We hope these efforts will be worked.

Please find attached a document with our responses.

Reviewer 2 Report

dear Authors,

This manuscript gives an excellent view of the planned purpose: analysis and quantification of the side effects after a booster dose of vaccination. In addition, this study would help additionally with the analysis of vaccination and booster hesitancy among the population.

The strategy developed by the authors for the study is well designed for accurate and quick performance of the survey. The nature of being a self-reported questionnaire filled online could introduce some inconveniences statistically, but the authors are pointing this possibility clearly in the text.

Data and its analysis are clearly structured and presented, helping the reader to have a good comprehension of the results. The discussion paragraph together with the conclusions are well exposed.

Summarizing, I am finding this work really interesting, well structured and presented in a clear manner, about the text, graphs and tables.

Author Response

First, many thanks are addressed to Reviewer #2 for the time dedicated to the review and the comprehensive, profound, and constructive remarks. The table below presents in detail how to address each comment; the references are to the final line numbers of the revised article. In addition, the added or changed text of the manuscript was marked using a yellow background. We believe that this paper will be highly cited by other authors.

Kindly find attached a document with our responses.

Reviewer 3 Report

How many people were vaccinated in Algiers during the epidemic?

How many received the inactivated vaccine and how may received the adenovirus vector vaccine?

With a sample size of just over 200, it might be difficult to generalise the side effects of each of the vaccines.

Is there any data on the attack rate or incidence of clinical and subclinical COVID 19 infection in Algeria in the various age groups?

Author Response

We would like to address our sincere gratitude to Reviewer #3 for the time dedicated to the review and the comprehensive, profound, and constructive remarks. The table below presents in detail how to address each comment; the references are to the final line numbers of the revised article. In addition, the added or changed text of the manuscript was marked using a yellow background. We believe that this paper will be highly cited by other authors.

Please find attached a document with our reponses.
